# INSERTANY3D: VLM-ASSISTED AND GEOMETRY-GROUNDED FRAMEWORK FOR 3D OBJECT INSERTION IN COMPLEX 3D SCENES

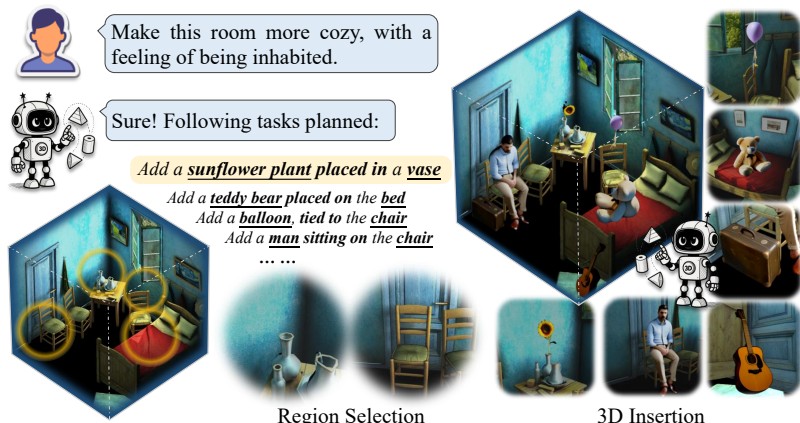

Figure 1: Example of our insertion effect. Our method can achieve perceptual insertion of complex 3D scenes driven by abstract users' intents, while ensuring both precise positioning and plausible interactions.

## ABSTRACT

The insertion of 3D objects into complex scenes is a critical task in 3D asset editing. Previous works use 2D inpainting models to edit multi-view images and lift them into 3D, which suffers from manual intervention and multi-view inconsistencies. To address these issues, we propose InsertAny3D, a novel framework for high-quality 3D object insertion guided by ambiguous natural language instructions in complex scenes. Our framework consists of two key components: (1) VLM-Assisted 3D Scene Understanding, which decomposes abstract user intents and selects optimal insertion regions through a hierarchical vision-language reasoning strategy; and (2) Geometry-Grounded 3D Object Insertion, which performs anchor-constrained 3D object generation and placement using depth-based feature matching and multi-view geometric verification to ensure spatial coherence. Extensive experiments demonstrate that InsertAny3D significantly outperforms existing methods in insertion precision, visual quality, and interactive usability.

## 1 INTRODUCTION

Insertion 3D objects into complex scenes plays a pivotal role in various industries, including game development, film production, and industrial design (Li et al., 2023a). This task involves not only placing objects accurately within a 3D environment but also ensuring that they integrate seamlessly with the surrounding elements. Achieving precise and intuitive 3D object insertion remains a significant challenge in the field, especially as the complexity of scenes increases.

Previous studies on 3D object insertion (Chen et al., 2024a; Ye et al., 2024; Cao et al., 2024) tackle this issue using a 2D-to-3D lifting approach. Their method involves rendering 2D background images from a 3D scene, inserting objects through a pre-trained inpainting model from multiple

viewpoints (Suvorov et al., 2021), and then reconstructing the 3D scene by lifting the edited 2D images back into 3D space. However, this approach has several key limitations: (1) It requires significant **manual intervention**. In complex environments especially when dealing with large and detailed scenes, users must manually select insertion points, define areas, and specify interaction methods, often leading to inaccuracies. This is due to the system's limited understanding of the 3D scene, which requires significant human intervention. (2) It suffers from **multi-view inconsistencies**, which leads to poor insertion quality. Since the method relies on multi-view image editing, maintaining spatial consistency across views is a major challenge. Additionally, iterative 3D lifting, which uses reconstruction loss, tends to accumulate errors and is computationally expensive. Consequently, the final outputs often exhibit poor detail, particularly at the interaction boundaries between inserted objects and the scene, where multi-view inconsistencies lead to blurred or imprecise edges.

To address these challenges, we introduce **InsertAny3D**, a framework that enables the seamless insertion of arbitrary 3D objects into complex scenes through ambiguous natural language instructions. This framework excels in handling scenarios with multiple interactive subjects, achieving precise object placement from simple, high-level commands. As shown in Fig. 1, InsertAny3D can perform accurate 3D object insertion, driven entirely by natural language prompts. This capability is powered by two key components:

To address the first challenge, we propose **VLM-Assisted 3D Scene Understanding**, which facilitates intent-driven planning and efficient region selection. Prior works (Cao et al., 2024; Chen et al., 2024a) in object insertion for complex scenes often struggles with abstract user instructions such as "Make this room more cozy" and requires significant manual effort. This challenge arises from the system's inability to interpret high-level intent in the context of complex 3D environments. Additionally, unlike prior methods (Cao et al., 2024) that rely on predefined regions, operating within such environments introduces the critical task of efficiently identifying the most suitable insertion areas from numerous potential candidates. To overcome these issues, we develop a novel VLM-based strategy, decomposing ambiguous user instructions into executable subtasks by leveraging reasoning capabilities. For region selection, we propose an optimized hierarchical method: instead of feeding all regions into the VLM, we first apply CLIP (Radford et al., 2021) for coarse filtering, followed by VLM for fine-grained selection, greatly enhancing efficiency.

After clearly defining the insertion task and corresponding regions, the second component, **Geometry-Grounded 3D Object Insertion**, achieves precise placement. We critically analyze previous 2D-to-3D lifting techniques (Haque et al., 2023; Cao et al., 2024) and highlight their shortcomings, particularly their vulnerability to multi-view inconsistencies. As a solution, we directly employ advanced 3D object generation models (Xiang et al., 2024) to generate the target 3D object and compute its alignment pose for accurate placement. However, this approach introduces new challenges, as converting reference images to 3D objects inevitably creates artifacts. To counter this, we use existing scene objects as anchors and co-generate them alongside the inserted object to improve consistency. For enhanced positional alignment accuracy, we conduct feature matching in depth space rather than in RGB space, thereby minimizing the influence of texture and lighting variations. Our method also includes a multi-view verification mechanism to resolve matching ambiguities and stabilize object alignment across multiple perspectives.

By applying our model to multiple scenes and conducting a comprehensive evaluation and comparison with other SOTA models, our method shows the superior capability of enabling the high-quality insertion of new objects into complex 3D interactive scenes from natural language prompt.

In summary, the main contributions of this paper are as follows:

• We introduce InsertAny3D, a novel framework for high-quality 3D object insertion into complex scenes, uniquely guided by ambiguous and high-level natural language instructions.

• We propose two key components in our framework. VLM-Assisted 3D Scene Understanding that enables effective planning and region selection through carefully designed efficient strategies and Geometry-Grounded 3D Object Insertion that introduces a novel insertion approach via anchor-constrained 3D object generation, incorporating robust depth-based grounding and multi-view geometry techniques to enhance ambiguity resolution.

• Extensive experimental results demonstrate the effectiveness of our method compared to previous approaches, achieving better detail preservation and higher insertion precision.

## 2 RELATED WORK

### 2.1 VISION LANGUAGE MODELS

Vision-Language Models (VLMs) have rapidly advanced as a powerful framework for integrating visual and linguistic information. The Vision Transformer (ViT (Dosovitskiy et al., 2020)) enabled scalable visual encoders, while models like CLIP (Radford et al., 2021) and ALIGN (Jia et al., 2021) leveraged contrastive learning on web-scale data to align modalities. This approach achieved strong zero-shot capabilities in tasks like classification, retrieval, and open-vocabulary recognition. Recent models such as BLIP-2 (Li et al., 2023b), Flamingo (Alayrac et al., 2022), and GPT-4V (OpenAI, 2023) further integrated vision with large language models, supporting multimodal reasoning, instruction following, and interactive tasks. These advances have established VLMs as key components in a wide range of cross-modal applications. As model capabilities have grown, researchers have begun exploring how to embed LLMs as "cognitive engines" to perform complex tasks beyond simple text generation. By integrating LLMs within a feedback loop, these models can not only "understand" a world described by text but also make decisions and execute actions to achieve a goal. For example, the ReAct framework (Yao et al., 2023) combines reasoning with action, allowing LLMs to think while operating. Furthermore, systems like AutoGPT (Yang et al., 2023) and BabyAGI (yoheinakajima, 2024) have demonstrated how LLMs can perform task decomposition, invoke external tools, and utilize self-feedback to accomplish intricate objectives. These advances showcase a shift towards empowering LLMs with more dynamic and interactive skills.

### 2.2 3D GENERATING MODELS

Recent advances in 3D generation, led by DreamFusion (Poole et al., 2022) have been largely driven by 2D priors such as Score Distilling Sampling (SDS) and Iterative Dataset Updating (IDU), enabling 3D asset synthesis from text or image prompts (Lin et al., 2023; Chen et al., 2023; Wang et al., 2023; Tang et al., 2023). In addition, multi-view diffusion techniques have significantly accelerated and enhanced image-to-3D generation (Liu et al., 2023b;a;c; Shi et al., 2023; Long et al., 2024; Chen et al., 2024b). In contrast, native 3D generation methods avoid multi-view inconsistencies by directly operating in the 3D domain or learning from 3D data. Early works such as Point-E (Nichol et al., 2022) and Shap-E (Jun & Nichol, 2023) explored generation via point clouds and implicit functions, while more recent methods like LRM (Hong et al., 2023) and DMV3D (Xu et al., 2023) leverage large-scale reconstruction models for fast and generalizable 3D synthesis. Further advances, including TextField3D (Huang et al., 2023), TRELLIS (Xiang et al., 2025), Sparc3D (Li et al., 2025), and Hunyuan3D 2.0 (Zhao et al., 2025), support open-vocabulary generation, multi-functional representations, high-resolution reconstruction, and texture synthesis. By operating directly in the 3D domain, these models gain an early and robust understanding of geometry and topology, offering clear advantages over 2D prior-based approaches in terms of geometric fidelity.

### 2.3 3D INSERTING MODELS

Recent successes of NeRF and 3D Gaussian Splatting have sparked growing interest in AI-driven methods for editable 3D scene generation, among which geometric object editing – such as inserting or modifying objects within complex environments – remains particularly challenging. Instruct-NeRF2NeRF (Haque et al., 2023) first demonstrated that DreamFusion's iterative optimization strategies can enhance the geometric consistency of 2D priors in 3D space. Follow-up work largely adopts this 2D prior-based paradigm for object-level editing (Weber et al., 2024; Abu-El-Haija et al., 2018; Liu et al., 2024), while some methods leverage fine-tuned multi-view diffusion models to bypass costly optimization (Wu et al., 2024a; Barda et al., 2025; Cao et al., 2024). Others employ local-to-global iterative refinement to improve object insertion (He et al., 2024; Zhuang et al., 2024). A separate line of work adopts coarse generation–insertion–optimization pipelines (Shahbazi et al., 2024; Chen et al., 2024a), which decouple object synthesis from scene context and rely heavily on manual input – such as view selection, mask drawing, and prompt design. Although these pipelines offer a structured workflow, their lack of interactive integration with the scene often leads to limited adaptability, suboptimal placement, and increased reliance on post-hoc refinement, making the process more fragile and less user-friendly. Despite these efforts, 2D prior-based approaches still rely on lifting 3D geometry from 2D supervision, making them prone to failure under atypical object-scene interactions – especially when such inconsistencies accumulate across multiple views.

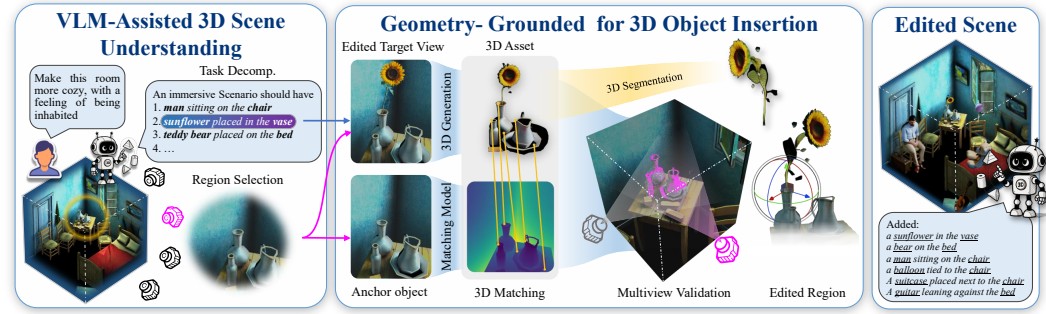

Figure 2: InsertAny3D achieves the insertion of arbitrary 3D objects into complex 3D scenes with ambiguous instructions. The pipeline begins with **VLM-assisted 3D Scene Understanding**, which identifies the optimal insertion region from a user's abstract instruction. An anchor view from this region is then fed into the **Geometry-Grounded for 3D Object Insertion** module. This module first creates a new, contextually integrated asset, and then uses depth-based feature matching and pose estimation to precisely ground the object in the scene's 3D geometry before insertion.

## 3    METHOD

Our method introduces a comprehensive framework for inserting objects into complex 3D scenes based on high-level user intent. As shown in Fig. 3, the process initiates with **VLM-Assisted 3D Scene Understanding**, where a VLM translates abstract users' instructions into concrete subtasks and a hierarchical strategy efficiently identifies the optimal region. Given the selected region, an anchor view is rendered to guide the next stage, **Geometry-Grounded 3D Object Insertion**, which produces a context-aware composite object aligned with both visual semantics and scene geometry. To accurately integrate the asset, we perform depth-based correspondence matching between the generated object and the original scene, enhanced by a multi-view verification strategy that resolves geometric ambiguities and ensures precise pose estimation. This modular design enables efficient, consistent insertion across diverse 3D scenes.

### 3.1    VLM-ASSISTED 3D SCENE UNDERSTANDING

As illustrated above, designing precise and machine-readable instructions in a complex scene is time-consuming and skill-intensive for users. Therefore, a critical prerequisite for inserting objects properly in a complex scene is to comprehend the entire scene and users' high-level, often abstract, instructions thoroughly. To address this, we propose a user-interactive system with dual functions: 1) Intent-Driven Planning: decomposing users' intent into multiple executable subtasks, 2) Efficient Region Selection: selecting the optimal region that fits users' needs.

**Intent-Driven Planning** To translate high-level and general abstract' intents into specific, executable subtasks, we employ a VLM to comprehend both the 3D scene and users' instructions. Specifically, as VLMs cannot directly process 3D information, our method first captures the scene context by rendering a set of images: 1) images rendered from the top four corners of the bounding box looking towards the center for global context, and 2) images rendered from random views for local details. The VLM then processes these images alongside users' instructions to identify potential object insertion tasks in the given 3D scene. For each identified task, it generates a structured prompt as a candidate for our generation pipeline. For instance, given the goal of "`make this room more cozy`"(as shown in Fig. 3), the VLM might propose inserting a sunflower into a vase and generate the corresponding: "`add a sunflower placed in the vase`". Through this process, we decompose a single abstract instruction into multiple machine-readable prompts, which are then executed sequentially.

Unlike previous methods designed for insertion, which are typically limited to a simple scene and a pre-defined region, the decomposed subtasks are conducted within a complex scene. As a complex scene is naturally composed of multiple simple regions, it raises a novel and critical challenge: how can we identify the optimal region for each subtask?

**Efficient Region Selector** To address the challenge of region selection, the system must locate a region within the complex scene that best aligns with the semantics. This requires a deep understanding of spatial relationships in the given 3D scene. A trivial yet computationally prohibitive method would be to render the scene from numerous viewpoints, feeding each resulting image into the VLM to identify the optimal one. The core flaw of this naive method lies in its inefficiency: the VLM is forced to process a vast number of regions that are semantically irrelevant to the sub-

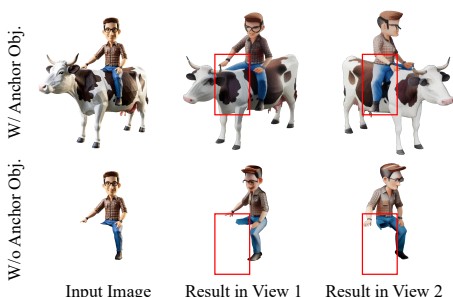

Figure 3: The necessity of demonstrating combination generation in the generation of interactive Assets. When the inserted object is partially obscured, the 3D generation model cannot complete the obscured part, resulting in errors.

task. To address this issue, we propose an efficient, hierarchical region selection strategy. Our strategy first employs a computationally lightweight coarse filter to rapidly discard irrelevant regions as a High-Recall Coarse Filter, followed by a more sophisticated fine filter to make the final selection. Specifically, we use CLIP (Radford et al., 2021) as the coarse filter due to its significantly lower computational overhead compared to a VLM. For each rendered image, we compute the cosine similarity between the CLIP image features and the text features of the subtask prompt. Regions with high similarity scores are retained as candidates. While CLIP-based filtering is effective at efficiently narrowing down the search space, it lacks the detailed 3D scene understanding required for valid anchor selection. Specifically, it is prone to false positives due to semantic ambiguity—for instance, erroneously matching a 'vase' prompt to a 2D painting of a vase rather than a physical object. Furthermore, it often fails to differentiate between accessible regions and those that are occluded or geometrically unsuitable. To address this, we employ a VLM as a fine-grained verifier. The VLM processes only the small set of candidate regions screened by CLIP, leveraging its superior reasoning capabilities to select the single optimal region that best satisfies the subtask.

Our hierarchical selection method significantly enhances filtering efficiency while preserving high performance in complex scenes. The primary source of this efficiency gain is the observation that in any given scene, the number of semantically relevant views constitutes only a small fraction of the total renderable images. By first isolating this small candidate pool, our method avoids exhaustive computation. Subsequently, the second-stage, fine-grained filtering by the VLM ensures that the truly optimal perspective from within this candidate set is selected, thus guaranteeing accuracy. This synergy ensures both rapid and accurate region selection, even within highly cluttered scenes.

## 3.2 GEOMETRY-GROUNDED 3D OBJECT INSERTION

**Anchor-Constrained 3D Asset Synthesis.** Prior methods for 3D scene editing often employ a two-stage paradigm: performing multi-view 2D inpainting, followed by 3D reconstruction. This approach, however, exhibits significant drawbacks. It is critically dependent on the 2D inpainting model's ability to maintain cross-view consistency, and the iterative process is not only time-consuming but also prone to error accumulation, frequently yielding models with geometric inaccuracies and textural artifacts. Consequently, this paradigm is ill-suited for robust and efficient 3D editing in complex scenes.

To overcome these limitations, we propose an anchor-constrained pipeline that generates a 3D asset by constraining it with its immediate interactive context. Our core strategy is to jointly synthesize the new object with a key contextual object, which we term the *anchor object*. This approach uses the anchor as a strong geometric and pose prior, resolving ambiguities inherent in generating from a single viewpoint. As shown in Fig. 3, a standard image-to-3D model may produce a malformed result from an occluded image of a person on a horse, or misinterpret the pose for a prompt like `a person sitting on a chair` without the chair's context.

Our pipeline directly addresses these challenges through a three-stage process. First, we perform 2D editing on a rendered image from a target region to create a composite that explicitly depicts the desired interaction. This composite image then guides a 3D generation model to output a single, uni-

fied 3DGS containing both the new asset and the anchor, enforcing the correct interactive pose. To isolate the newly generated asset, we employ an adapted version of the text-driven 3D segmentation pipeline, SAGS (Ververas et al., 2024). Specifically, we integrate LangSAM (Medeiros et al., 2023) as the 2D front-end to enable open-vocabulary text prompting and apply a stricter multi-view voting threshold to suppress inconsistent segmentation masks. This approach advantageously avoids a full reconstruction pipeline.

**Robust Depth-Based Grounding.** A critical step in our pipeline is to accurately register the generated asset back into the original scene. A naive approach using 2D image-based feature matching on the anchor object is fragile. This is because 2D editing and subsequent 3D generation inevitably introduce texture, lighting, and subtle geometric inconsistencies between the anchor in the scene and its counterpart in the asset. Due to the non-linear nature of back-projection, even minor 2D matching errors are amplified into significant inaccuracies in the final 3D pose.

To circumvent this fragility, we propose a robust registration strategy that operates directly on depth images, leveraging geometric consistency while avoiding the challenges of point cloud registration. The process is as follows: First, to eliminate interference from background features during matching, we employ LangSAM (Medeiros et al., 2023) for foreground object segmentation. Second, we render a depth map of the original scene from the selected viewpoint, denoted as $D_{\text{scene}}$. Then, we render a corresponding depth map from the same viewpoint for our synthesized 3D asset, denoted as $D_{\text{asset}}$. Finally, We employ an image feature matcher (Shen et al., 2024) directly on these two depth images, $D_{\text{scene}}$ and $D_{\text{asset}}$, to establish a set of dense and reliable 2D correspondences. Since each pixel correspondence $(u_s, v_s) \leftrightarrow (u_a, v_a)$ has an associated depth value from its respective map, we can lift each pair to a 3D-to-3D correspondence in camera space. This set of robust 3D correspondences allows us to solve for the rigid transformation (i.e., translation, rotation and an isotropic scaling ratio) that accurately aligns the asset to the scene. This approach is resilient to photometric variations and directly computes the initial pose from geometrically consistent matches using the Umeyama algorithm, bypassing the error amplification inherent in color-based matching.

**Disambiguating Matches with Multiview Geometry.** A fundamental challenge in feature matching is the ambiguity arising from symmetric or repetitive geometric structures, which leads to incorrect correspondences in a single-view context. Even when operating on depth map to mitigate texture-based ambiguity, specifying a user's selection among repeating instances remains non-trivial. Thus, we introduce a multi-view verification scheme that leverages depth parallax. By incorporating the distinct viewpoint, the parallax between the target object and other similar instances is amplified, enabling us to filter out ambiguous matches that fail to maintain geometric consistency across views.

Our matching model initially produces thousands of candidate point correspondences. Let $\mathcal{P}_1 = \{(p_1^i, p_2^i)\}_{i=1}^{N_1}$ be the set of matching point pairs from the primary view, where $p_1^i$ is a point in the original scene and $p_2^i$ is its corresponding point on the generated asset. Similarly, let $\mathcal{P}_2 = \{(q_1^j, q_2^j)\}_{j=1}^{N_2}$ be the set of matches from an auxiliary view.

We define a spatial proximity function with a 3D threshold $\delta$, where $\delta$ represents the Euclidean distance in 3D space:

$$\mathcal{D}_\delta(x, y) = \begin{cases} 1, & \text{if } \|x - y\|_2 < \delta \\ 0, & \text{otherwise} \end{cases} \tag{1}$$

For each pair $(p_1^i, p_2^i) \in \mathcal{P}_1$, define its neighbor set in the side view as:

$$\mathcal{J}_i = \left\{ j \,\middle|\, \mathcal{D}_\delta(p_1^i, q_1^j) = 1 \right\} \tag{2}$$

Then, the match $(p_1^i, p_2^i)$ is considered valid only if:

$$\mathcal{J}_i \neq \emptyset \quad \text{and} \quad \forall j \in \mathcal{J}_i, \ \mathcal{D}_\delta(p_2^i, q_2^j) = 1 \tag{3}$$

The final verified set of correspondences is defined as:

$$\mathcal{P}_{\text{valid}} = \left\{ (p_1^i, p_2^i) \in \mathcal{P}_1 \,\middle|\, \mathcal{J}_i \neq \emptyset \text{ and } \forall j \in \mathcal{J}_i, \ \mathcal{D}_\delta(p_2^i, q_2^j) = 1 \right\} \tag{4}$$

This strategy enforces cross-view geometric consistency, retaining only those correspondences from the primary view that find geometrically coherent support in the auxiliary view. It effectively disambiguates matches for repetitive structures by ensuring that a valid match holds true from multiple perspectives.

Table 1: Quantitative comparison and user study results. Our method significantly outperforms previous baselines on both automatic metrics (HPSv2, VLM Judge) and human preference evaluations.

| Method | HPSv2 ↑ | VLM Judge ↑ | | | Human Preference ↑ | | |
|---|---|---|---|---|---|---|---|
| | | Visual | Rational | Geometry | Aesthetic | Precision | Overall |
| Gaussian Editor | 0.257 | 1.63 | 2.30 | 3.08 | 10% | 6% | 7% |
| Gaussian Grouping | 0.264 | 7.30 | 7.73 | 7.39 | 31% | 28% | 20% |
| MVInpainter | 0.258 | 4.27 | 4.75 | 4.48 | 8% | 5% | 8% |
| InsertAny3D(Ours) | **0.266** | **8.14** | **8.67** | **8.42** | **51%** | **61%** | **65%** |

## 4 EXPERIMENTS

### 4.1 EXPERIMENT DETAILS

**Implementation** We adopt GPT-4o (OpenAI, 2024) as the underlying model for the Agent to handle visual understanding and complex reasoning tasks. For 2D editing, we utilize Fooocus (lllyasviel, 2025), and TRELLIS is employed for 3D generation. We use the optimized SAGS for 3D segmentation, and GIM (Shen et al., 2024) is applied to achieve depth map matching. For multi-view matching, we set 0.05 (in the unit of "meter" in Unity) as the threshold for the original scene. Except for GPT-4o, which is accessed via API, all other models can run on a single NVIDIA GeForce RTX 3090.

**Datasets** Since there are no open-source datasets or benchmarks for 3D objects insertion, we manually collect multiple large scenes from (Sketchfab, 2025) to form a dataset. This dataset contains diverse scenes, including farms, bedrooms and so on. We conducted all experiments on this dataset.

**Evaluation Metrics** Due to the lack of well-established metrics for 3D insertion, we use two complementary metrics. HPSv2 (Wu et al., 2023)—trained on human judgments—assesses alignment with the task prompt by evaluating text-to-image aesthetic appeal. Drawing on GPTEval3D (Wu et al., 2024b)—a metric originally designed for 3D generation tasks—we adapt and tailor it for 3D insertion scenarios, resulting in our VLM metrics. These metrics focus on visual and geometric quality across three dimensions: visual quality, generation rationality, and insertion pose accuracy. By inputting multi-view images of original scenes and comparative outputs, the VLM evaluates these dimensions via customized prompts, enabling human-like preference assessment of both visual and geometric aspects. Detailed prompts are provided in the supplementary materials.

### 4.2 EXPERIMENTAL RESULTS

**Quantitative Analysis.** To quantitatively evaluate our approach, we compare it with three representative baselines: GaussianEditor (Chen et al., 2024a), GaussianGrouping (Ye et al., 2024), and MVInpainter (Cao et al., 2024). GaussianEditor relies on naive depth estimation, which performs reasonably on planar surfaces but fails in complex geometries. GaussianGrouping requires manual placement and lacks seamless interaction with the scene. MVInpainter suffers from error accumulation during multi-view propagation, often leading to distorted and inconsistent results. As shown in Tab. 1, our method achieves state-of-the-art performance across all VLM metrics, with notable improvements in both visual fidelity and geometric fitting. More importantly, it effectively resolves the interaction between inserted objects and the original scene, while robustly preventing common failures such as penetration and floating artifacts. In terms of text–image alignment, our approach further obtains a competitive HPSv2 score of 0.266, demonstrating its ability to faithfully capture and render user intent.

In terms of efficiency, our method is quicker than previous works that are based on 2D editing, taking less than 2 minutes for one subtask when tested on a 3090 GPU. This duration includes a 30s planing and region selection, 10s 2D editing phase, 20s for 3D generation, 30s for 3D segmentation, and 15s for GIM. By comparison, MVInpainter requires approximately 60 seconds for the 2D editing phase alone—and during the 3D training phase, its duration can range from 5 to 10 minutes depending on parameter settings.

**Qualitative Analysis.** We provide a qualitative comparison to visually demonstrate the superiority of our method. Fig. 4, showcases editing results from different methods on a variety of scenes and prompts. These examples highlight our method's ability to generate objects that are not only

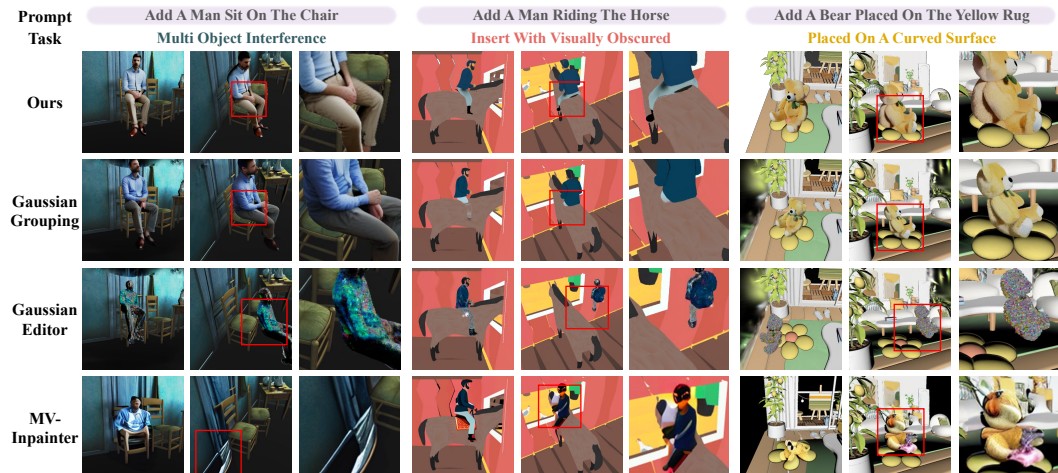

Figure 4: **Qualitative Results of the Comparative Experiment.** In various complex scenarios, Our Approach has achieved better quality in interaction processing, including the degree of fit between the generated object's geometry and the original scene, the degree of fit between the inserted pose, and visual quality, and is highly consistent with the input text description.

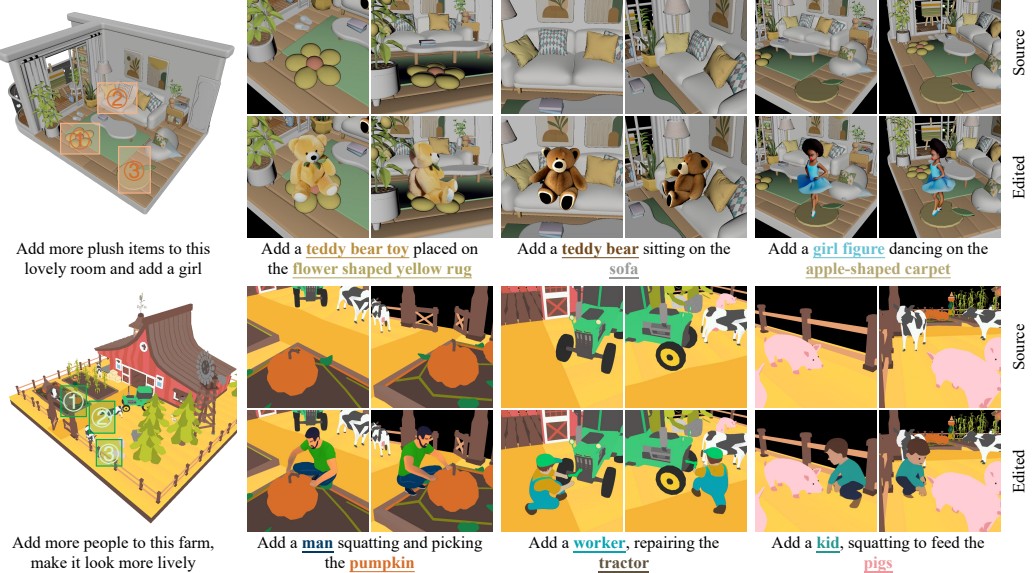

Figure 5: **Qualitative examples of our scene editing framework.** Given user instructions (left), our method generates semantically grounded object insertions that interact naturally with the 3D environment. **Top block:** In a cozy indoor scene, the model successfully adds multiple plush toys and a dancing girl on a specified carpet, while maintaining spatial consistency. **Bottom block:** In a farm scene, our method introduces new characters (a man, worker, and child) performing context-aware actions (e.g., picking pumpkins, repairing a tractor, feeding pigs), enhancing the scene's liveliness and interactivity. Each row shows the original view (top) and our edited result (bottom).

semantically correct but also harmoniously integrated into the original scene. We also show the effects of our method in different scenarios in Fig. 5. Experiments show that our method can perform high-quality insertion in complex scenes.

**User Study** To evaluate the perceptual quality of our generated results, we conducted a user study focusing on two key aspects: Aesthetic and Pose Precision. Each aspect captures a different dimension of output quality from the human perspective. The average scores are summarized in Tab. 1. Our method achieves significantly higher ratings across all three aspects compared to baseline approaches. Qualitative feedback from users further supports these findings, highlighting our model's ability to maintain both realism and semantic faithfulness in complex scenes.

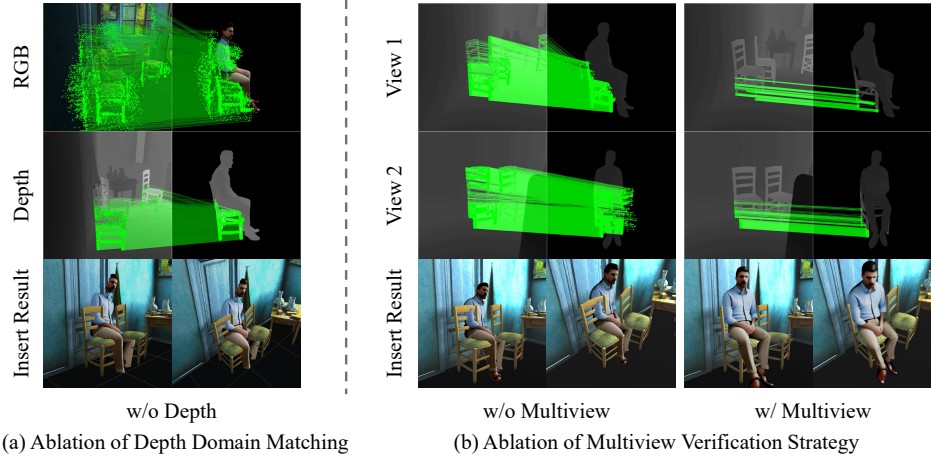

(a) Ablation of Depth Domain Matching (b) Ablation of Multiview Verification Strategy

Figure 6: **Qualitative Results of the Ablation Study.** (a) Comparison between depth-based and RGB-based matching. RGB matching introduces noisy correspondences due to inconsistent image edits, while depth-based matching captures the underlying geometry and produces cleaner results. (b) Effect of multiview validation. In ambiguous cases with repeated structures, the absence of multiview leads to incorrect one-to-many correspondences. With multiview, geometric consistency across views disambiguates matches and improves overall alignment accuracy.

Table 2: Ablation experiments of our proposed method. We evaluale our method without depth(Dep.), multi-view(MV), and SAM. Ablation experiments show that each component of our method is highly effective.

| Method | HPSv2 ↑ | VLM Judge ↑ | | |
| --- | --- | --- | --- | --- |
| | | Visual | Rational | Geometry |
| w/o Dep. | 0.265 | 1.95 | 1.53 | 1.61 |
| w/o SAM and Multiview | 0.262 | 3.28 | 3.00 | 1.92 |
| w/o Multiview | 0.256 | 4.70 | 4.81 | 3.95 |
| InsertAny3D(full method) | **0.266** | **5.83** | **6.19** | **5.06** |

**Ablation Experiments** To demonstrate the effectiveness of depth-based grounding and multiview disambiguation, we conduct an ablation study on our Unity dataset. As some cases completely failed in the ablation experiment, we excluded them from the experiment and recalculated the quantitative metrics of the full method on reamined cases. Our method comprises three main components: LangSAM preprocessing, depth-based matching for initial correspondences, and multiview validation (MV) for final refinement. We evaluate three settings: (1) replacing the depth input with RGB; (2) removing both LangSAM and multiview; and (3) removing multiview only. As LangSAM serves as an essential prerequisite for the operation of MV, we do not conduct ablation experiments on MV alone. Instead, we compare the performance of LangSAM with that of LangSAM+MV to demonstrate the necessity of multiview validation. Quantitative comparisons are shown in Tab. 2, and qualitative examples are illustrated in Fig. 6. The depth-based input significantly outperforms RGB matching, validating its robustness against inconsistent image editing. Moreover, removing MV leads to a clear performance drop, confirming its role in resolving ambiguous matches, especially in scenes with repetitive structures.

## 5 CONCLUSION

In this paper, we present InsertAny3D, a framework that addresses challenges of user interaction and quality in 3D object insertion. Departing from prior 2D-lifting methods, our approach uniquely combines a VLM-powered assistant for intuitive, language-driven task planning with a novel, 3D-native anchor-guided insertion technique. By operating directly in 3D and leveraging scene anchors with depth-space feature matching, our method ensures high-fidelity results and coherent object integration. Experiments demonstrate that InsertAny3D offers an efficient and robust solution for high-quality object insertion into complex scenes from natural language prompts.

ETHICS STATEMENT

We confirm that our work adheres to the ICLR Code of Ethics (https://iclr.cc/public/CodeOfEthics). The research involves no human subjects or sensitive data, utilizing only publicly available datasets that comply with their respective usage terms. Our 3D insertion framework is designed for creative and design purposes, with no intended application in harmful contexts. We confirm strict adherence to research integrity principles, including honest data reporting, transparent methodology documentation, and avoidance of any form of scientific misconduct, and the study design complies with standard academic ethics guidelines. We remain committed to addressing any ethical concerns raised during the review process.

REPRODUCIBILITY STATEMENT

We have made every effort to ensure the reproducibility of our experiments. All experiments were designed to be executable on a single 3090 GPU, and multiple trials were conducted to mitigate randomness. The specific experimental configurations and the prompts used for the VLM metrics are detailed in the appendix, enabling independent replication of our results.

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

# A APPENDIX

## A.1 THE USE OF LARGE LANGUAGE MODELS

During the research process, large language models (LLMs) were solely employed for text polishing and proofreading. We conducted thorough checks to ensure the accuracy of the text, strictly avoiding any fabrication or misrepresentation. Importantly, LLMs were not involved in any other aspects of the research, including research ideation, experimental design, data analysis, or result interpretation, to maintain the integrity of the scientific process.

## A.2 IMPLEMENTATION DETAILS

### A.2.1 REGION SELECTION

In this study, a Region is defined as an area in the world coordinate system with xyz coordinates and a radius of d. We perform grid sampling over the scene's bounding box, sampling 10 points along the length/width dimensions and 5 along the height dimension. When selecting a Region, the camera is aimed at the center of the Region. The pose with an Euclidean distance of d, a pitch angle of $\theta$, and a rotation angle of $\gamma$ is taken as the main view. Three camera view images are formed by rotating $\pm 48°$ around the rotation angle (for example, when $\gamma = 0°$, the views are $-48°$, $0°$, and $+48°$) to cover the main observation angles of the Region as much as possible, serving as the basis for Region screening and the reference view for Multiview Validation. Here, d is valued from 1 to 5 with a precision of 0.1 (Unity unit); the pitch angles are $10°$, $20°$, $30°$, and $40°$ for downward views, and the rotation angles are integer multiples of $12°$ for convenient engineering implementation. During rendering, the fov is $53.13°$, and the rendering resolution is $1024 \times 1024$.

### A.2.2 DATASET DETAILS

To benchmark InsertAny3D comprehensively, we constructed a diverse evaluation dataset sourced from Sketchfab, specifically designed to challenge 3D insertion capabilities across varying scales and complexities.

- Scale: The dataset consists of 20 distinct 3D scenes.
- Instruction Set: For each scene, we utilized a VLM to analyze the context and generate 5 potential insertion commands, resulting in a total of 100 specific insertion tasks used for evaluation.
- Diversity: To ensure broad coverage, the dataset spans indoor and outdoor environments, includes CG, painterly, and photorealistic styles, contains tasks involving planar insertion, curved-surface insertion, and human-object interaction–based insertion, providing a thorough stress test for generalization and robustness.

### A.2.3 METRICS DETAILS

We use HPSv2 as the task matching metric, directly matching the task prompt with multi-view rendered images to compute the consistency between the insertion result and the target. For visual quality, insertion rationality, and geometric accuracy, we employ a Visual-Language Model (VLM) with the prompt specified in the attached "evaluation_prompt.txt". The VLM prompt is designed to ensure the model fully understands the editing task during metric output, focusing the quality assessment on the insertion rather than the background.

For each VLM evaluation, we input three rendered images of the original scene and the comparative method, using views selected in the Region Selection stage. Both metrics are evaluated three times with the same views to mitigate random errors from pre-trained model assessments.

For each user study case, participants are presented with randomly ordered results (all rendered multi-view images) from our method and baseline methods, and are asked to select the superior one. The specific definitions provided to the participants were: 1) Aesthetic: Visual fidelity, lighting harmony, and the absence of artifacts. 2) Pose Precision: Geometric accuracy (e.g., checking for floating objects, collisions, or incorrect orientations). 3) Overall Quality: A holistic assessment of which result best satisfies the user instruction while maintaining realism

In the experiments, MVInpainter's pipeline requires multi-view mask propagation, which is not always reliable; additionally, GaussianEditor suffers from excessive pose calculation errors, leading to failure in locating target objects, hence causing missing values in some metrics. When computing average metrics, we ignored these failure cases. Even so, our method achieves better performance.

## A.3 VIDEO ATTACHMENT DESCRIPTION

Our supplementary materials include a video that showcasesthe visual effects of our method in detail through multiperspective filming. In various complex environments, ourmethod has achieved the best level in terms of generationquality, object fit, and insertion accuracy.

## A.4 DETAILED INDICATOR RESULTS AND VISUAL EFFECTS

- Figure 7, Figure 8, Figure 9 illustrates three scenarios used in our comparative experiments, demonstrating the superiority of our method in terms of stability, prompt consistency, as well as visual and geometric quality.

- Figure 10 provides additional results of other scene showcasing the generalizability of our method.

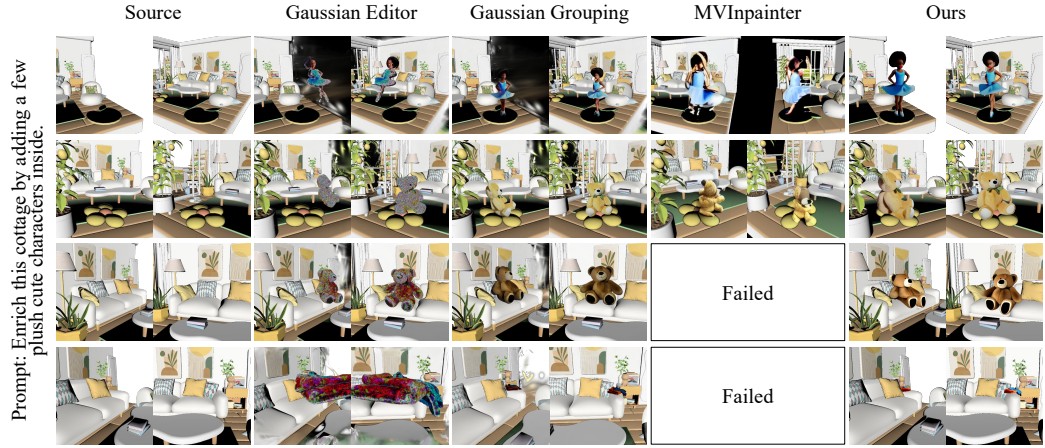

Figure 7: Gallery of comparative experiments results 1/3.

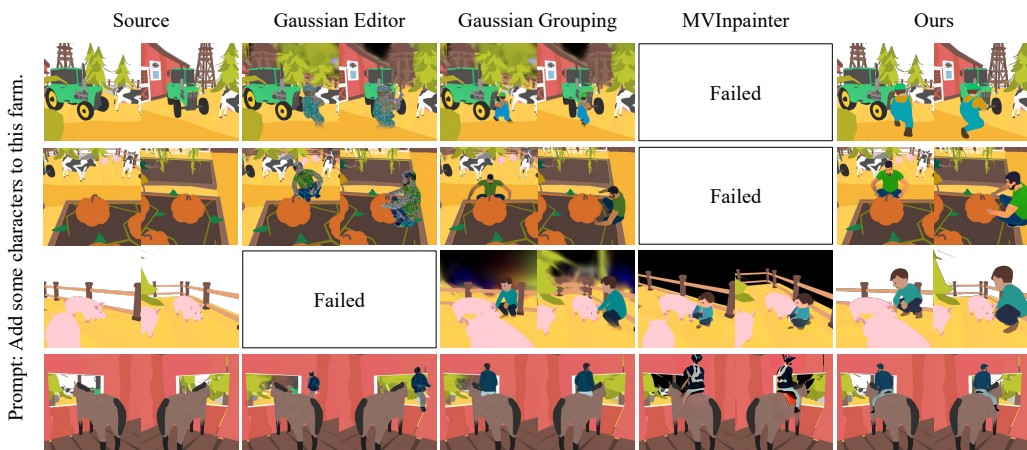

Figure 8: Gallery of comparative experiments results 2/3.

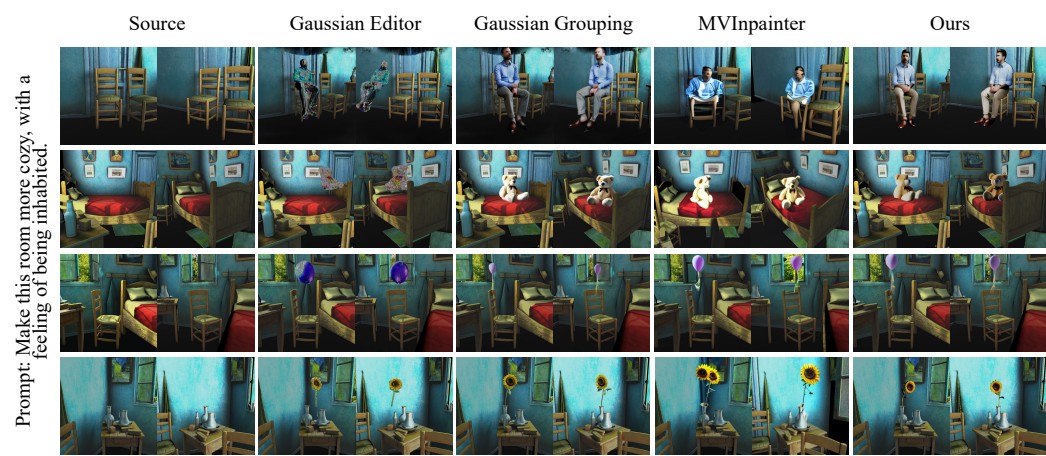

Figure 9: Gallery of comparative experiments results 3/3.

Prompt: Add some transportation and neon lights to this city.

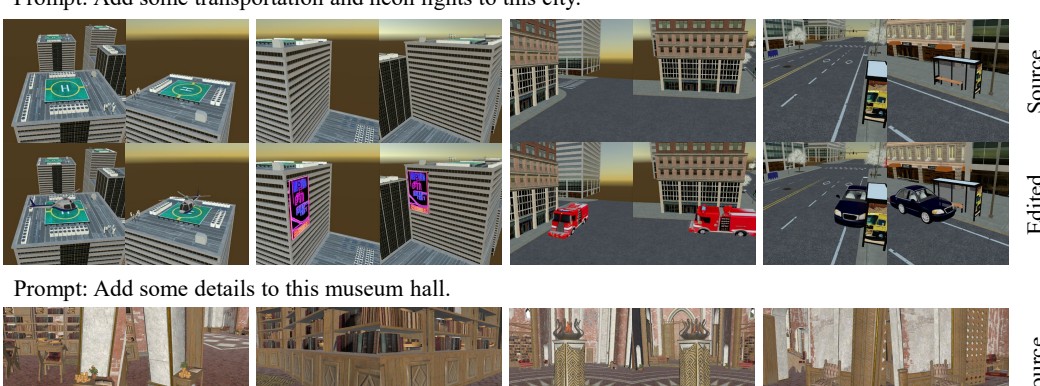

Prompt: Add some details to this museum hall.

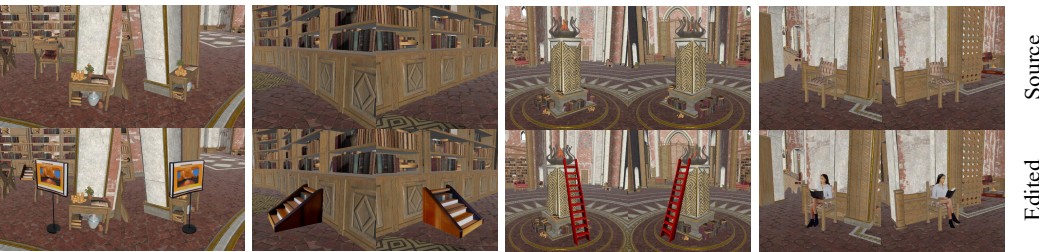

Figure 10: Additional 3D object insertion effects using our algorithm. The examples illustrate our method's ability to accurately insert objects on planes and precisely position them in scenes with complex interactions. Our algorithm achieves consistent accuracy and realistic interaction across scenes of different scales and styles.

## A.5 ANALYSIS OF UNOPTIMIZED NAIVE INSERTION TASKS DECOMPOSITION

To clearly demonstrate the necessity and comparative effectiveness of our task decomposition optimization, we conducted two distinct experiments using the Visual Language Model (VLM):

1. **Naive Execution:** The VLM was prompted using the original, unoptimized list of insertion tasks ("The initial list of insertion tasks").

2. **Optimized Execution:** The VLM was prompted using our refined task list, which was decomposed and optimized based on our defined criteria.

The comparative results are illustrated in the Fig. 11: The **left side** presents the initial input image and the complete VLM input prompt text. The **right side** displays the generated outputs from the Naive Execution and the Optimized Execution, respectively.

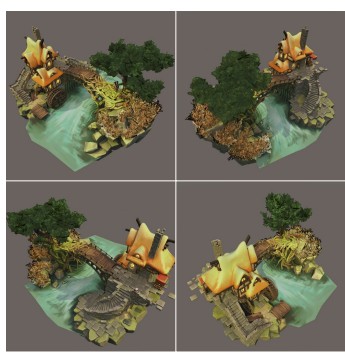

**Prompt: Bring some life to this small town**

**a) Naive task decomposition**

**Insert Moss:** Add patches of moss onto the rocks by the riverside.
**Insert a Boat:** Place a small rowboat in the water.
**Insert a Barrel:** Place a large wooden barrel on the stone platform area.
**Insert a Bench:** Place a bench on the ground facing towards the house.
**Insert a Lantern:** Place a lantern on the bridge.
**Insert a Cat:** Place a small ginger tabby cat sitting on the wooden bridge, looking down at the water.
**Insert Sacks:** Stack two or three burlap sacks, presumably filled with flour, on the small wooden dock near the water level.
**Insert a Scarecrow:** Place a scarecrow amongst the dried grass bushes near the tree.
**Insert a Character:** Add a baker attendant character at the bakery serving counter.
**Insert a Birdhouse:** Hang a wooden birdhouse from a branch of the large tree.

**b) Our task decomposition**

**Stack Firewood:** Insert a stack of firewood logs against the exterior stone wall next to the main wooden door of the bakery.
**Place Sacks:** Place several burlap sacks on the ground adjacent to the base of the large waterwheel.
**Place Basket:** Place a bamboo basket on the wooden planks of the small lower dock.
**Place Easel:** Place an easel with signage upright on the stone pavement beside the bakery display counter.
**Insert Lilies:** Insert water lilies and pads floating on the water surface surrounding the large submerged rocks.
**Mount Lantern:** Mount a lantern fixture onto a main vertical wooden support post of the bridge railing.
**Place Walking Cat:** Place a cat in a walking pose on the wooden plank surface of the arched bridge.
**Mount Weathervane:** Mount a weathervane atop the highest ridgeline point of the main thatched roof.
**Stack Sacks:** Stack three **full, heavy-looking burlap sacks** stably on the wooden planks of the small lower dock.
**Hang Birdcage:** Hang a birdcage from an overhanging branch of the large tree.

Figure 11: Comparative results of VLM generation: (a) Input Image and Prompt; (b) Result from Naive Task Decomposition vs. Result from Our Task Decomposition.

The initial list of insertion tasks failed to meet our optimization criteria concerning correctness of interaction, environmental adaptability, and task feasibility. We categorize the inconsistencies into three core areas:

**Clarity of expression** Ours instructions are clearly more coherent and execution-ready than the naïve ones. They use consistent action verbs, maintain uniform structural patterns, and avoid overly specific or ambiguous descriptions. In contrast, the naïve instructions mix object types, levels of detail, and grammatical styles, making them less stable as scene-editing directives.

**Conflicting Task Definition and Functionality** This category covers operations that violate the task scope or scene logic. The *Insert Moss* task is a texture editing operation, not an Object Insertion task. The *Insert a Lantern* task proposes placing the object on the bridge surface, which is functionally implausible; lanterns must be mounted or suspended. The *Insert Sacks* description is flawed by including an invisible and speculative attribute ("presumably filled with flour"), which is irrelevant to the VLM's generation capability and introduces ambiguity. Specifically, "presumably filled with flour" is an unnecessary and unclear addendum, as it describes the content rather than the form. This ambiguous instruction risks the VLM failing to capture the intended visual material state and perceived weight of the object.

**Missing or Unrealistic Interaction and Pose Detail** These errors relate to poor definition of an object's posture or interaction. The combined action for *Insert a Cat* ("sitting" and "looking down") results in an unnatural or physically strained pose, compromising the correctness of interaction. Similarly, the *Insert a Character* task lacks a defined action or specific posture, preventing the generation of a figure that plausibly interacts with the serving counter.

**Ambiguous Feasibility and Placement** This final category addresses insufficient precision in defining the insertion location, which impacts visibility. The instruction for *Insert a Barrel* uses the vague location "on the stone platform area." This ambiguous placement risks the barrel being occluded or placed in a secondary view, violating the task feasibility constraint that the inserted object must be clearly visible.

