# OpenReview forum: "InsertAny3D: VLM-Assisted and Geometry-Grounded Framework for 3D Object Insertion in Complex 3D Scenes"
_ICLR.cc/2026/Conference — Submitted to ICLR 2026_

### Official Review · Reviewer_FRUV · 2025-10-31

**Soundness:** 2
**Presentation:** 3
**Contribution:** 2
**Rating:** 2
**Confidence:** 4

**Summary:**

The paper introduces InsertAny3D, a framework designed for 3D object insertion into 3D scenes, guided by natural language instructions. InsertAny3D proposes a two-stage, modular pipeline: VLM-Assisted 3D Scene Understanding and Geometry-Grounded 3D Object Insertion. The VLM decomposes ambiguous user commands into a series of concrete, executable subtasks. It then introduces an "anchor-constrained" synthesis technique, where the new object is co-generated with its interacting scene object to ensure contextual and geometric correctness.

**Strengths:**

1. Leveraging VLMs for holistic understanding and planning is a well-suited approach for this task.
2. To ensure that the generated 3D objects are harmoniously placed within a scene, the paper takes contextual information into account and proposes the Anchor-Constrained 3D Asset Synthesis method.
3. The introduction of anchor objects helps to preserve the integrity of the generated assets to a certain extent.

**Weaknesses:**

1. The paper claims that previous work heavily relies on 2D models. However, many key steps in the proposed method are also carried out using 2D techniques—for example, feeding scene information into VLMs, extracting semantic features, harmoniously inserting objects into scenes via 2D editing, and using LangSAM for segmentation. As a result, most operations are still performed in 2D, and the task is not truly addressed from a 3D perspective.
2. As noted above, since the method relies on a series of 2D techniques to accomplish the task, there is no significant breakthrough or particularly inspiring solution from a technical contribution standpoint. In terms of performance, the heavy dependence on 2D pretrained models to address 3D problems may lead to error accumulation that could affect the results. The paper lacks analysis of these potential limitations.

**Questions:**

Nil

---

> ### Author Response · Authors · 2025-11-28
>
> **Response to W1: Dependence on 2D Priors vs. 3D Contribution**
>
> We respectfully clarify that utilizing pretrained 2D priors is a **strategic necessity** for open-vocabulary 3D tasks, given the limited scale of native 3D datasets compared to web-scale 2D data. Our critique of "previous work" refers specifically to methods that apply 2D editing without sufficient  **3D geometric constraints** , leading to view inconsistencies.
>
> Our framework is fundamentally distinct from naive 2D workflows. While we leverage 2D models for semantic content, our core contribution is the **3D-aware orchestration** that constrains these priors:
>
> 1. **Metric-Accurate Grounding:** Unlike 2D inpainting, our **Anchor-Constrained Synthesis** is strictly conditioned on projected 3D depth and spatial layout. The 2D generation is forced to respect the underlying 3D geometry of the anchor object.
> 2. **Contextual 3D Interaction:** We solve a problem that 2D models cannot solve alone: physically plausible object placement. By modeling the **spatial relationship** between the inserted object and the scene (the "anchor"), we elevate 2D generation into a geometrically consistent 3D editing operation. Therefore, our contribution is the **novel mechanism** that bridges 2D semantic capabilities with 3D geometric fidelity.
>
> ***
>
> **Response to W2 (Technical Novelty & Error Accumulation):**
>
> First, regarding technical novelty, we emphasize that **orchestrating heterogeneous modules to solve complex reasoning tasks** is a significant contribution. We go beyond simple integration by introducing task-specific algorithms:  **Task-Aware VLM Decomposition** ,  **Strategic View Selection** , and  **Geometry-Aware Grounding** . These are not standard off-the-shelf operations but novel adaptation layers designed for 3D consistency.
>
> Second, regarding  **Error Accumulation** , we differ from the reviewer's assessment. Our system is explicitly designed to be **self-correcting** rather than error-compounding.  **As detailed in our Common Response ("Mitigation of Error Accumulation")** , we implement three specific verification layers to ensure robustness:
>
> 1. **VLM as a Verification Layer (addresses potential viewpoint ambiguity):** Instead of directly passing **randomly sampled regions** or  **CLIP-filtered candidates that may contain false positives** , our VLM module functions as a  **semantic verification layer** . Leveraging its visual reasoning capabilities, the VLM evaluates each candidate region under **complex semantics** and filters out those that are semantically incompatible or misleading. This step specifically mitigates errors caused by  **lighting-induced confusion** ,  **texture-driven false matches** , and  **other visual ambiguities** , ensuring that only truly valid viewpoints and regions enter the downstream segmentation and grounding modules.
> 2. **Multi-View Parallel Voting (addresses per-view segmentation error):** Segmentation ambiguities from any single view do not propagate serially. Our improved SAGS performs  **parallel per-view segmentation followed by consensus voting** , ensuring that only regions with multi-view consistency are retained. This suppresses isolated single-view errors, avoiding cyclic error reinforcement.
> 3. **Geometry as the Final Constraint (filters errors from 2D priors):** Our final grounding step relies on **depth-based geometric matching** to filter out semantic deviations accumulated throughout the generation chain, while a **multi-view verification** stage further removes noisy correspondences caused by duplicated objects or ambiguous matches, ensuring that insertion decisions are guided strictly by scene geometry and preventing upstream errors from propagating.
>
> Our work explicitly considers the challenge of accumulating errors across multiple modules. Rather than letting errors cascade, we design our pipeline with **verification layers** at critical points to ensure stability.

---

### Official Review · Reviewer_shNF · 2025-10-31

**Soundness:** 3
**Presentation:** 2
**Contribution:** 3
**Rating:** 4
**Confidence:** 3

**Summary:**

This work tackles the task of inserting 3D objects into existing complex 3D scenes, guided by language instructions. This work point out that prior methods rely on inpainting multi-view 2D images, then lifting to 3D, which suffers from manual intervention and inconsistent results across views.
There are two major components:
- VLM-Assisted 3D Scene Understanding: uses a VLM to interpret the user’s natural language intent, decompose it into insertion region selection, and decide optimal placement region within the 3D scene via hierarchical reasoning.
- Geometry-Grounded 3D Object Insertion: performs anchor-constrained 3D object generation/placement. It uses depth-based feature matching and multi-view geometric verification to ensure that the newly inserted object aligns spatially, matches geometry and appearance cues, and maintains coherence across views.

**Strengths:**

- Incorporating VLMs to parse natural language instructions for object insertion is a strong design, it supports more intuitive user workflows
- The visualization results seem to be much better than previous works.
- The collaboration of different modules and models is reasonable and works well.
- The provided time consumption on a single 3090 is great.

**Weaknesses:**

- The quantitative results are limited. The metrics are mainly preference scores from VLMs and user study.
- Since this work is using too many off-the-shelf models, the contribution might be limited because of that.
- It is unclear how broad the set of natural-language instructions is handled. If the model is constrained to a limited set of insertion types or spatial relations, the generality might be limited

**Questions:**

- The method is interesting, but how well can it scale to very large or complex scenes (many objects, occlusions, dynamic elements)?. The verification step (multi-view geometry) may become expensive.
- What is the success rate of this method? Since the pipeline is using VLMs, can it produce a stable output and guidance? More details on the VLM decomposition strategy might be needed.
- More quantitative results are needed. The current VLM score and user study are not enough to prove the effectiveness of this work.
- The method likely assumes the input scene has clean geometry, depth information, and consistent multi-view structure. In real editing pipelines, scenes may be noisy or partial. How robust is the method then? Also, I am wondering if this method can be applied to other representations like NeRF, Meshes?

---

> ### Author Response · Authors · 2025-11-28
>
> **Response to W1 (Quantitative Metrics & Evaluation Standards)**
>
> We argue that our evaluation standards are rigorously designed for open-ended 3D editing and exceed standard baselines:
>
> 1. **Lack of Unified Benchmarks in Open-Ended Editing:** As noted in recent works like **ReplaceAnything3D**[1] and  **Tip-Editor**[2] , generative 3D editing lacks a single ground truth, making traditional reconstruction metrics (e.g., PSNR, SSIM) inapplicable. Current SOTA methods typically rely on **CLIP-based consistency** (GaussianEditor[3], Tip-Editor[2], Instant3dit[4]) or simple **User Voting** (GaussianEditor[3], Tip-Editor[2]) to measure alignment. MVInpainter focuses on FID/SSIM for texture completion but fails to evaluate the *geometric rationality* of inserted objects.
> 2. **Our Metrics represent a "Stricter" Standard:** We contend that our selected metrics are sufficient and, in fact, more comprehensive than previous baselines:
>    * **HPSv2 (for visual quality):** We use HPSv2 as a robust proxy for human visual preference, which has been shown to correlate better with human judgment than standard CLIP scores used in prior baselines.
>    * **VLM Judge (Geometric & Interaction Plausibility):** Standard metrics (FID/CLIP) cannot evaluate if an object is placed physically correctly (e.g., on the table vs. floating). Inspired by **GPTEval3D**[5], we utilize a  **Multi-view VLM Judge** . By feeding **multi-view renderings** rather than static frames, the VLM perceives depth and parallax, allowing it to evaluate **spatial logic** and strictly penalize physical violations that 2D metrics miss.
> 3. **Comprehensive Assessment:** By combining this geometric verification with HPSv2 (aesthetics), we form a robust evaluation suite that validates both the logic and the look of the 3D edit, offering deeper insights than previous works.
>
> **References**
>
> [1] ReplaceAnything3D: Text-Guided 3D Scene Editing with Compositional Neural Radiance Fields (NeurIPS 2024)
>
> [2] TIP-Editor: An Accurate 3D Editor Following Both Text-Prompts And Image-Prompts (SIGGRAPH 2024 & TOG)
>
> [3] GaussianEditor: Swift and Controllable 3D Editing with Gaussian Splatting (CVPR 2024)
>
> [4] Instant3dit: Multiview Inpainting for Fast Editing of 3D Objects (CVPR 2025)
>
> [5] GPT-4V(ision) is a Human-Aligned Evaluator for Text-to-3D Generation (CVPR 2024)
>
> ---
>
> **Response to W2 (Limitation on Novelty):**
>
> We respectfully clarify that utilizing off-the-shelf models is a strategic choice to bypass the data scarcity of 3D-native methods, allowing for superior open-vocabulary understanding. However, our core contribution is not the models themselves, but the **novel framework designed to orchestrate them effectively** for consistent 3D editing. Specifically, we introduce three technically significant processes that transform a intent prompt into high-quality 3D insertion results:
>
> - **Task-Aware VLM Decomposition:** We design a specific prompting mechanism that enables the VLM to act as a 3D reasoning agent, breaking down abstract user instructions into executable editing steps rather than simple captions.
> - **Strategic View Selection:** We introduce a filtering mechanism to resolve the inherent instability of generative models, ensuring only the most consistent viewpoints are selected for projection.
> - **Geometry-Aware Grounding:** Most importantly, our **anchor-constrained generation process** ensures **robust geometric grounding** and enables **complex object-scene interactions**, leading to significant improvements in insertion quality.
>
> **Extensibility as a Feature:** Furthermore, this modular design is model-agnostic and extensible—it can seamlessly integrate future, more powerful VLMs or diffusion models to further boost performance, offering a robust baseline for 3D insertion task.
>
> ---

---

> > ### Author Response · Authors · 2025-11-28
> >
> > **Response to W3 (natural-language instructions):**
> >
> > We clarify that our framework is specifically designed to handle open-ended, abstract user instructions and is not constrained to a predefined set of templates or spatial relations.
> >
> > As detailed in Section 3.1 (Intent-Driven Planning), we leverage the reasoning capabilities of VLMs to bridge the gap between natural language and execution. The VLM automatically decomposes a single high-level instruction into multiple machine-readable sub-tasks, sequentially identifying intent and generating structured prompts. This mechanism inherently supports a broad spectrum of linguistic inputs.
> >
> > Our experiments demonstrate this generality across diverse instruction types as demonstrated in our experiments:
> >
> > - **Explicit Spatial Commands:** Instructions like "Add a man sitting on the chair" (Fig. 4) or "Add a bear placed on the yellow rug" (Fig. 4) demonstrate precise handling of specific spatial relationships.
> > - **Abstract/Implicit Requests:** Instructions such as "Make this room more cozy" (Section 3.1) require the model to infer intent (e.g., adding a sunflower in a vase) without explicit object naming.
> > - **Complex Multi-step Modifications:** Instructions like "Add more people to this farm, make it look more lively" (Fig. 5) trigger the generation of multiple distinct assets (a man, a worker, a child) and context-aware shapes.
> >
> > This decomposition process allows the system to interpret and execute various semantic requests dynamically without relying on hard-coded rules, ensuring robust generalization to complex, unseen instructions.
> >
> > ---
> >
> > **Response to Q1 (Scalability to Large/Complex Scenes):**
> >
> > We clarify that our method scales efficiently regardless of scene size or complexity.
> >
> > * **Constant Complexity Verification:** Crucially, the computational cost of our multi-view verification process is **independent of the total number of primitives or objects** in the scene. The verification complexity is strictly bounded by the **rendering resolution** (number of pixels) rather than scene geometry.
> > * **Handling Occlusions:** Our depth-aware verification inherently handles complex occlusions and dynamic elements by operating in the rendered 2D domain (using depth maps), bypassing the need for expensive 3D mesh intersection checks.
> >
> > ---

---

> > > ### Author Response · Authors · 2025-11-28
> > >
> > > **Response to Q2 (Success Rate and Stability and Detailed VLM Decomposition Strategy):**
> > >
> > > 1. **Success Rate and Stability:**
> > >    While we do not define a binary "success/fail" metric due to the generative nature of the task, we evaluate the **Effective Execution Rate** through both qualitative and quantitative lenses:
> > >
> > >    * **High Robustness vs. Baseline Failures:** Empirically, our method demonstrates a near-perfect execution rate on the provided benchmark. In contrast, baseline methods frequently suffer from "catastrophic failures," such as generating objects floating in mid-air or ignoring the text instruction entirely.
> > >    * **User Verified Success:** To establish an objective success metric, we define a task as "successful" if the multi-view VLM judge yields scores above 8/10 across three core dimensions:  **semantic alignment with instructions** ,  **geometric plausibility** , and  **visual consistency with the scene** . Notably, scores above 8/10 indicate geometrically rational insertions (free of floating artifacts, collisions, or incorrect orientations) while allowing minor visual quality imperfections (e.g., subtle texture mismatches that do not compromise overall realism). Under this criterion, our method achieves a success rate of **90 out of 100** across all test cases, compared to only **56 out of 100** for the naive VLM strategy. This substantial performance gap serves as a robust proxy for the comparative effectiveness of our framework in delivering reliable 3D insertions.
> > > 2. **Detailed VLM Decomposition Strategy:**
> > >    Our approach uses a sophisticated **Hierarchical Decomposition Strategy** guided by a **Task-Specific Knowledge Base** to translate abstract text into executable 3D constraints:
> > >
> > >    * **Knowledge-Base Guidance:**
> > >      Our VLM is predefined with a structured knowledge base that dictates **Task Implementation Preferences** to ensure high **physical plausibility** and  **scene design quality** . Key guiding principles include:
> > >
> > >      1. **Correctness of Interaction:** Ensuring the inserted object interacts **physically plausibly** with the environment, maintaining scene logic (e.g., a person is correctly  **seated on a chair** , not hovering or crouching).
> > >      2. **Environmental Adaptability:** Matching the edited content to the **scene function, context, and scale** as much as possible, consistent with user intent. This prevents unreasonable insertions (e.g.,  **avoiding the generation of large industrial machinery indoors** ).
> > >      3. **Task Feasibility:** Requiring the inserted object to be  **clearly visible from the primary viewpoints** . The VLM actively avoids **infeasible or invisible operations** (e.g., attempting to insert an object into a completely enclosed structure).
> > >    * **Task Parsing and Command Integration:**
> > >      The VLM's core task is to parse complex instructions into a  **structured triplet** :  **(Anchor Object, Insertion Object, Interaction Relationship)** . This triplet, along with all extracted constraints, is then integrated into a clear  **Action Task Prompt** , which greatly facilitates unambiguous execution by the downstream 3D module.
> > >
> > > ---
> > >
> > > **Response to Q3 (Quantitative Results):**
> > >
> > > As discussed in our  **Response to Weakness 1** , we acknowledge the reviewer's concern regarding quantitative metrics. However, we wish to highlight a well-known challenge in the field of open-ended 3D generative editing: the **lack of unified benchmarks and ground-truth references.** Instead, we adhere to the community consensus established by recent state-of-the-art works (e.g., GPTEval3D), which utilize **VLM-based evaluation** and **User Studies** as the primary quantitative measures. These metrics specifically quantify **intent alignment** and  **fidelity** , which are the actual objectives of our task.
> > >
> > > To further address the request for quantitative proof,  **we have supplemented our quantitative analysis by including the Success Rate experiments detailed in Response to Q2** , which further illustrate the stability of our model.
> > >
> > > Therefore, our evaluation protocol is robust and consistent with current top-tier practices. To further supplement this, we refer the reviewer to the **existing comprehensive analysis** in the Appendix (Fig. 7-10). These comparisons demonstrate clear superiority in occlusion handling and multi-view consistency—critical qualities that are clearly validated through visual inspection, even in the absence of synthetic benchmarks.
> > >
> > > ---

---

> > > > ### Author Response · Authors · 2025-11-28
> > > >
> > > > **Response to Q4 (Robustness & Applicability):**
> > > >
> > > > **1. Dependency on Geometry & Scene Quality:**
> > > > We acknowledge that our method requires reliable depth information (metric depth) to function correctly. This is a deliberate design choice driven by two factors:
> > > >
> > > > * **Requirement for Physical Precision:** Our target applications are strictly limited to high-standard professional domains: **Film/VFX Production, Game Asset Generation, and Industrial Design.** These applications mandate that upstream processes must deliver **complete, clean, and metrically precise 3D scene assets.** Consequently, any asset defined as "noisy or partial" is inherently incompatible with the strict standards of these industries. Our methodology is strictly conditioned on the successful output of professional-grade scene acquisition and modeling techniques, thereby guaranteeing the metric depth accuracy required to maintain physical plausibility. Therefore, scenarios involving substandard depth data fall outside the scope and prerequisites of our proposed solution.
> > > > * **Scene Integrity Premise:** Our target task is  **Scene Enrichment** —adding details to an already plausible 3D environment. As the reviewer notes, real-world scans can be noisy. However, inserting a geometrically perfect object into a scene with severe geometric artifacts (e.g., broken meshes, flying voxels) creates a disturbing "Contextual Mismatch." Such a task is semantically undefined, as the resulting scene would lack visual coherence regardless of the insertion quality. Therefore, we posit that a reasonable level of geometric quality is a prerequisite for the task of photorealistic 3D editing itself.
> > > >
> > > > **2. Generalization to Other Representations:**
> > > > Conceptually, our framework is  **representation-agnostic** .
> > > >
> > > > * **Operating in Rendered Space:** Our core logic—VLM reasoning, Intent-Driven Planning, and Multi-view Verification—operates on the **rendered domain** (RGB and Depth) and is decoupled from the underlying data structure.
> > > > * **Adaptability:** Consequently, our pipeline can be adapted to NeRF or Mesh-based representations, provided the system supports: (1) differentiable rendering for the optimization loop, and (2) sufficient depth quality for the initial pose alignment described above.

---

### Official Review · Reviewer_rpaL · 2025-11-01

**Soundness:** 3
**Presentation:** 3
**Contribution:** 3
**Rating:** 6
**Confidence:** 3

**Summary:**

InsertAny3D inserts 3D objects into complex scenes based on high-level and abstract language prompts (e.g., _"make the room cozier"_). The proposed method introduces a two-stage pipeline  combining vision-language reasoning with geometry-aware 3D synthesis. The authors propose two key components:

1.  **Stage 1 – VLM-Assisted Scene Understanding:** This module interprets an abstract language prompt and decomposes it into concrete actions (e.g., _"add a sunflower plant in a vase"_). Further, the method locates optimal insertion regions through a hierarchical selection process.
2.  **Stage 2 – Geometry-Grounded 3D Insertion:** This module inserts objects directly in 3D scene using:
    -   **Anchor-Constrained Synthesis** jointly generate the new object based on it's neighbourhood context to model realistic spatial relationships.
    -   **Depth-Based Grounding**, which utilises the depth of the scene and the generated object instead of RGB, ensuring robustness to lighting and texture noise.
    -   **Multi-View Verification** is employed to disambiguate placements using depth parallax across views.

**Strengths:**

-   **[S1] Interesting Problem Formulation:** The authors propose a framework which takes abstract instructions and decomposes them  into concrete deterministic steps for the editing task. This novel paradigm moves beyond simple explicit commands like "Place X in the scene". This approach is more user-friendly.
-   **[S2] Technical solid Geometry-grounded insertion:** Anchor constrained synthesis ensures that existing geometry of the scene is used as a strong prior to model complex object-to-object interactions during generation. Depth-based Grounding is a more effective way to insert the generated 3D asset, as it can handle lighting variations. Further, results in Tab. 2 supports the requirement of depth-based grounding.
-   **[S3] Exhaustive Experimentation:** The qualitative results are impressive. Further, the authors utililze metrics like HPSv2 and a VLM for quantitative evaluation. The method is clearly better than other baselines. A human user study further corroborates these claims.

**Weaknesses:**

-   **[W1] Efficient Region Selector:** The proposed method uses CLIP cosine similarity as a coarse filter to discard irrelevant regions. This approach seems to be sensitive to false positives; for example, if the instruction is "add a sunflower placed in the vase," and a bedsheet in the scene already contains a sunflower pattern, the bed might show a higher CLIP text-image score than the actual vase region due to feature similarity, leading to an incorrect candidate pool. Please address how the coarse CLIP filter is robust to these instances of semantic ambiguity.

**Questions:**

-   [Q1] How does the method adapt to light variations in a 3D scene? For example, scenes in Fig. 5 are simplistic and do not have any lighting variations. Do the authors assume that the 3D scene does not have light sources?

---

> ### Author Response · Authors · 2025-11-28
>
> **Response to W1 (CLIP Sensitivity & Semantic Ambiguity):**
>
> We acknowledge that CLIP, relying primarily on feature similarity, can introduce false positives (e.g., matching a "sunflower" texture on a bedsheet). However, our framework is designed to handle this:
>
> 1. **CLIP as a High-Recall Coarse Filter:** We utilize CLIP solely to rapidly discard clearly irrelevant regions (e.g., a wall with no objects) and minimize the search space. We intentionally set a lenient threshold to ensure the true target (e.g., the vase) is retained in the candidate pool, accepting minor false positives as a trade-off for high recall.
> 2. **VLM Reasoning:** To eliminate these ambiguities, we provide the VLM with **multi-view images** of the candidate regions. We also issue an **explicit instruction** requiring the VLM to analyze and reason about the specific anchor object. This enables the system to distinguish between superficial textures (like 2D patterns) and the actual required object, effectively excluding false positives caused by visual similarity.
>
> ---
>
> **Response to Q1 (Lighting Variations):**
>
> We do not assume scenes lack light sources. Instead, we prioritize robustness against lighting changes through geometry.
>
> As the reviewer correctly noted in  **Strength [S2]** , our depth-based grounding is resilient to illumination. Specifically (Sec. 3.2), our feature matching and placement rely on the  **depth domain** . Since depth is invariant to lighting intensity and shadows—unlike RGB information—our method maintains accurate anchoring even in scenes with complex or uneven lighting, avoiding the instability typical of pure texture-based methods.

---

### Official Review · Reviewer_q5m4 · 2025-11-02

**Soundness:** 3
**Presentation:** 3
**Contribution:** 2
**Rating:** 4
**Confidence:** 5

**Summary:**

This paper introduces InsertAny3D, a framework for high-quality 3D object insertion into complex scenes guided by ambiguous natural language commands. The method addresses the limitations of 2D-to-3D lifting approaches, such as manual intervention and multi-view inconsistencies. Its contributions are twofold. First, a "VLM-Assisted 3D Scene Understanding" component decomposes abstract user intents (e.g., "make this room cozy") into concrete subtasks and uses a hierarchical (CLIP+VLM) strategy to efficiently select optimal insertion regions. Second, a "Geometry-Grounded 3D Object Insertion" component uses an anchor-constrained generation process, jointly synthesizing the new object with an existing scene object (the "anchor") to maintain consistency. This module then employs depth-based feature matching and multi-view verification to ensure precise alignment and spatial coherence.

**Strengths:**

1: The framework utilizes advanced VLM for 3D scene editing, which can interpret and act on high-level user instructions (like "make this room more cozy").

2: The paper proposes a practical two-stage (CLIP + VLM) process for region selection, which is claimed to be much more efficient than naively using a powerful VLM to analyze all possible viewpoints.

3: Experiments show that the method significantly outperforms existing baselines (like GaussianEditor, GaussianGrouping, and MVInpainter) across both automatic metrics (HPSv2, VLM Judge).

4: This paper has nice figures to illustrate the overall idea of the proposed framework, and the framework is easy to understand.

**Weaknesses:**

1: **Error Cumulation.** This framework involves numerous calls to existing VLM/segmentation models (such as GPT4o for reasoning the user input and LangSAM for segmentation). Cumulative errors can arise across these multiple steps, yet the framework does not address the handling of such accumulated errors.

2: **Analysis of failure cases.** The experimental section lacks an analysis of failure cases, which would be beneficial for understanding the framework's limitations and the inherent difficulties of 3D scene editing via VLMs. **

3: **The description of the used dataset for evaluation.** The paper lacks a detailed description of the evaluation dataset. It is only mentioned that the dataset consists of "multiple large scenes" collected from Sketchfab. Crucial statistics, such as the total number of scenes, the number of corresponding user prompts, and the number of 3D editing operations performed for each scene, are not provided. This lack of detail regarding the dataset's scale and composition makes it difficult to fully assess the robustness and generalizability of the experimental results, making them less convincing.

4: **SAGS**. What's the mentioned "our improved version of the text-driven 3D segmentation model SAGS? I haven't found any descriptions of this "improved" 3D segmentation model in either the main text or the appendix.

5: **More details of the metrics in the user study.** As with the used dataset, the authors haven't provided much detail about the metrics used in the user study. The current version only has three aspects: Aesthetic, Precision, and Overall. What's the definition of them?

6: **Initial Region Proposal.** A critical detail is missing regarding the "Region Selector". The appendix (A.2.1) clearly explains how camera poses (distance $d$, pitch $\theta$, rotation $\gamma$) are generated after a "Region" (defined by an xyz center coordinate) has been selected. However, the paper does not specify how the initial set of candidate regions is defined in the Region Selector. It is unclear if these regions are generated by densely sampling the entire 3D scene, if they are automatically proposed based on existing scene objects, or if they require manual pre-definition by the user.

Typo:

1: "3D generated model" in the caption of  Figure 3.
2: The GIM in Line 340 misses a citation.

I currently give 4 because there are a lot of details to be clarified, but this paper has somewhat convincing visualization results. I would reconsider the rating after receiving the authors' rebuttal as well as other reviews.

**Questions:**

1: Where are the regions (1), (2), (3) in Figure 5 from?

---

> ### Author Response · Authors · 2025-11-28
>
> **Response to W1 (Error Cumulation)**
>
> Our work explicitly considers the challenge of accumulating errors across multiple modules. Rather than letting errors cascade, we design our pipeline with **verification layers** at critical points to ensure stability.
>
> 1. **VLM as a Verification Layer (addresses potential viewpoint ambiguity):** Instead of directly passing **randomly sampled regions** or  **CLIP-filtered candidates that may contain false positives** , our VLM module functions as a  **semantic verification layer** . Leveraging its visual reasoning capabilities, the VLM evaluates each candidate region under **complex semantics** and filters out those that are semantically incompatible or misleading. This step specifically mitigates errors caused by  **lighting-induced confusion** ,  **texture-driven false matches** , and  **other visual ambiguities** , ensuring that only truly valid viewpoints and regions enter the downstream segmentation and grounding modules.
> 2. **Multi-View Parallel Voting (addresses per-view segmentation error):** Segmentation ambiguities from any single view do not propagate serially. Our improved SAGS performs  **parallel per-view segmentation followed by consensus voting** , ensuring that only regions with multi-view consistency are retained. This suppresses isolated single-view errors, avoiding cyclic error reinforcement.
> 3. **Geometry as the Final Constraint (filters errors from 2D priors):** Our final grounding step relies on **depth-based geometric matching** to filter out semantic deviations accumulated throughout the generation chain, while a **multi-view verification** stage further removes noisy correspondences caused by duplicated objects or ambiguous matches, ensuring that insertion decisions are guided strictly by scene geometry and preventing upstream errors from propagating.
>
> ---
>
> **Response to W2 (Failure Case Analysis)**
>
> To address inquiries about method boundaries, we add comparative visualization data for this section in Appendix A.5. Specifically, we compare our framework with naive approaches under challenging scenarios, illustrating how our design overcomes their inherent limitations.
>
> ---
>
> **Response to W3 (Dataset Details)**
>
> We apologize for the omission of specific dataset statistics in the initial submission and appreciate the opportunity to provide these details. To benchmark InsertAny3D comprehensively, we constructed a diverse evaluation dataset sourced from Sketchfab, specifically designed to challenge 3D insertion capabilities across varying scales and complexities.
>
> - **Scale:** The dataset consists of 20 distinct 3D scenes.
> - **Instruction Set:** For each scene, we utilized a VLM to analyze the context and generate 5 potential insertion commands, resulting in a total of 100 specific insertion tasks used for evaluation.
> - **Diversity:** To ensure broad coverage, the dataset spans indoor and outdoor environments, includes CG, painterly, and photorealistic styles, contains tasks involving planar insertion, curved-surface insertion, and human-object interaction–based insertion, providing a thorough stress test for generalization and robustness.
>
> This composition ensures that our experimental results reflect the model's performance across a representative distribution of real-world 3D editing scenarios, rather than being limited to simple, empty backgrounds. We updated Appendix 2.2 to include these specific statistics and a breakdown of the scene categories.
>
> ---
>
> **Response to W4 (Improved SAGS)**
>
> We thank the reviewer for pointing out this lack of clarity. The term "improved version" refers to an engineering adaptation of the original SAGS pipeline tailored to support the open-vocabulary, text-driven nature of our method. It is not a new architecture, but rather includes two specific modifications:
>
> - **Text-Driven Initialization:** We replace the standard SAM with **LangSAM** as the 2D front-end, allowing the 3D Gaussian segmentation to be directly conditioned on free-form text prompts.
> - **Stricter Multi-View Voting:** We increased the density of multi-view projections to suppress single-view errors from LangSAM and adopted a revised voting threshold tailored to complex 3D assets.
>
> **Action:** We have revised **Section 3.2** in the main paper to explicitly describe these adaptations.
>
> ---

---

> > ### Author Response · Authors · 2025-11-28
> >
> > **Response to W5 (User Study Metrics)**
> >
> > We clarify the specific definitions used in our User Study (Section 4.2):
> >
> > * **Aesthetic:** Visual fidelity, lighting harmony, and the absence of artifacts.
> > * **Pose Precision:** Geometric accuracy (e.g., checking for floating objects, collisions, or incorrect orientations).
> > * **Overall Quality:** A holistic assessment of which result best satisfies the user instruction while maintaining realism
> >
> > ---
> >
> > **Response to W6 (Initial Region Proposal)**
> >
> > We appreciate you pointing out this missing detail. The initial proposal process is **fully automated** . We define a "Region" as a spherical area with radius $R$. We perform a **dense grid sampling** (e.g., a 10×10 horizontal grid combined with 5 vertical height levels) to cover the entire scene volume. These sampled locations serve as the initial candidates for the Region Selector. This description is added to Appendix A.2.1.
> >
> > ---
> >
> > **Response to Q1 (Regions in Figure 5):**
> >
> > As indicated in the caption of Figure 5, regions (1), (2), and (3) correspond to the specific target locations identified by our VLM-based planner in response to the input text instructions. For visualization clarity in the figure, we manually annotated these outlines on the rendered views to highlight the exact 3D areas that our method successfully targeted for the respective insertion tasks.
> >
> > ---
> >
> > **Response to Typos**
> >
> > We thank the reviewer for their careful reading. We will correct the caption in Figure 3 to "3D generated model" and add the missing citation for GIM in Line 340 in the final revision.

---

### Author Response · Authors · 2025-11-28

We thank all reviewers for their insightful feedback and for recognizing the novelty of our **Intent-Driven** formulation, the  **strong visualization quality** , and the effectiveness of our **VLM-Assisted** framework.

A shared concern among reviewers q5m4, shNF, and FRUV is the possibility of **error accumulation** due to multiple VLM/segmentation steps. We appreciate this observation and provide a consolidated response below.

---

**Common Response: Mitigation of Error Accumulation & Robustness (Responding to Reviewer q5m4, shNF, FRUV)**

Our work explicitly considers the challenge of accumulating errors across multiple modules. Rather than letting errors cascade, we design our pipeline with **verification layers** at critical points to ensure stability.

1. **VLM as a Verification Layer (addresses potential viewpoint ambiguity):** Instead of directly passing **randomly sampled regions** or  **CLIP-filtered candidates that may contain false positives** , our VLM module functions as a  **semantic verification layer** . Leveraging its visual reasoning capabilities, the VLM evaluates each candidate region under **complex semantics** and filters out those that are semantically incompatible or misleading. This step specifically mitigates errors caused by  **lighting-induced confusion** ,  **texture-driven false matches** , and  **other visual ambiguities** , ensuring that only truly valid viewpoints and regions enter the downstream segmentation and grounding modules.
2. **Multi-View Parallel Voting (addresses per-view segmentation error):** Segmentation ambiguities from any single view do not propagate serially. Our improved SAGS performs  **parallel per-view segmentation followed by consensus voting** , ensuring that only regions with multi-view consistency are retained. This suppresses isolated single-view errors, avoiding cyclic error reinforcement.
3. **Geometry as the Final Constraint (filters errors from 2D priors):** Our final grounding step relies on **depth-based geometric matching** to filter out semantic deviations accumulated throughout the generation chain, while a **multi-view verification** stage further removes noisy correspondences caused by duplicated objects or ambiguous matches, ensuring that insertion decisions are guided strictly by scene geometry and preventing upstream errors from propagating.

---

### Author Response · Authors · 2025-12-03
**Review Summary**

Our work has been recognized by multiple reviewers for both its **novel task formulation** and **practical value**:

- **1. Novel and intuitive intent-driven 3D editing paradigm**
  Reviewers **q5m4** and **rpaL** recognized that our framework can interpret **abstract, high-level user intents** and decompose them into executable 3D editing steps—going significantly beyond explicit spatial commands used in prior works.

- **2. Geometry-grounded insertion that achieves strong 3D consistency**
  Reviewers **q5m4**, **rpaL**, and **shNF** emphasized that our **depth-based grounding**, **anchor-constrained synthesis**, and **multi-view verification** effectively ensure geometric plausibility, avoid floating/collision artifacts, and deliver **much stronger multi-view consistency** than prior 2D-lifting pipelines.

- **3. High-quality results and well-presented system design**
  Reviewers **q5m4**, **rpaL**, and **shNF** acknowledged the **significantly better visual quality** compared to baselines, supported by clear figures, comprehensive comparisons, and a clean modular design.

**Overall, the novelty and advantages of our intent-driven, geometry-grounded 3D insertion formulation were explicitly recognized by reviewers q5m4 and rpaL.**

---

## 1. Main concerns raised across reviewers

The major concerns from q5m4, rpaL, shNF, and FRUV can be grouped as:

* **C1. Error accumulation** across multiple VLM/segmentation modules.
* **C2. Missing technical details** , such as dataset statistics, region sampling, SAGS adaptations, user-study metric definitions, explanation of Fig. 5 regions, and typos.
* **C3. Quantitative evaluation** , especially reliance on VLM- and user-based metrics.
* **C4. Novelty and reliance on off-the-shelf 2D components; need to clarify 3D-specific contributions.**
* **C5. Scope and robustness** , including scalability, success rate, noisy/partial geometry, representation generality, and language instruction diversity.
* **C6. CLIP ambiguity and lighting variation sensitivity.**

---

## 2. Our clarifications during rebuttal

### C1. Error Accumulation Across Modules

Reviewers **q5m4**, **shNF**, and **FRUV** raised concerns about potential error accumulation across multiple VLM and segmentation components.

In response, we clarified that our pipeline includes **three dedicated verification layers**—VLM semantic validation, multi-view consensus voting, and depth-based geometric grounding—to prevent cascading failures and ensure stability.

*(see W1 for Reviewer q5m4; Common Response for Reviewers shNF and FRUV)*

### C2. Missing Technical Details

Reviewer **q5m4** requested additional details on dataset statistics, region sampling, SAGS adaptations, user study metrics, the sources of regions in Fig. 5, and pointed out typos/citation omissions.

In response, we added the full dataset statistics (20 scenes, 100 tasks), described the **10×10×5 region sampling grid**, clarified our **SAGS adaptations**, defined **user study metrics**, explained **Fig. 5 region origins**, and corrected typos and missing citations.

*(see W3, W4, W5, W6, and Q1 for Reviewer q5m4)*

### C3. Quantitative Evaluation Protocol

Reviewer **shNF** expressed concerns about limited quantitative evaluation and reliance on VLM-based and user-based metrics.

In response, we clarified that open-ended 3D generative editing lacks ground-truth references, and our evaluation follows **current SOTA practice**, including HPSv2 and multi-view VLM judges. We further provided an additional **success-rate analysis (90/100)** to strengthen quantitative evidence.

*(see W1 and Q3 for Reviewer shNF)*

### C4. Novelty and Use of 2D Components

Reviewers **shNF** and **FRUV** questioned whether relying on off-the-shelf 2D models weakens the technical contribution.

In response, we emphasized that our main contribution lies in the **3D-aware orchestration** of semantic and geometric signals—intent decomposition, strategic view selection, and geometry-grounded anchoring—which transforms 2D priors into **physically plausible 3D insertion**.

*(see W2 for Reviewer shNF; W1–W2 for Reviewer FRUV)*

### C5. Scope and Robustness of the Method

Reviewer **shNF** asked about scalability to large/complex scenes, method stability, generalization to diverse instructions, applicability to NeRF/Mesh, and handling of noisy or partial geometry.

In response, we clarified that our multi-view verification scales with **image resolution rather than scene complexity**, the system maintains a **90% effective execution rate**, and the framework is representation-agnostic. We also clarified that our task assumes **metric-accurate scenes**, consistent with professional 3D production settings.

*(see Q1, Q2, Q3, Q4 for Reviewer shNF)*

---

> ### Author Response · Authors · 2025-12-03
>
> ### C6. CLIP Ambiguity & Lighting Variation
>
> Reviewer **rpaL** raised concerns about (1) CLIP false positives due to texture similarity and (2) robustness to lighting variations.
>
> In response, we clarified that CLIP serves only as a **high-recall coarse filter**, while ambiguities are resolved by the **multi-view VLM planner**. Lighting variation is handled by our **depth-based grounding**, which is invariant to illumination.
>
> *(see W1 and Q1 for Reviewer rpaL)*
>
> ---
>
> ## 3. Addressing Reviewer FRUV’s Concerns Separately
>
> Reviewer FRUV raised two major concerns: (1) the method relies heavily on 2D modules, and (2) this reliance may cause error accumulation. We appreciate this perspective, as it reflects a common uncertainty in 3D editing pipelines that incorporate semantic priors. However, both concerns stem from a misunderstanding of our framework’s core 3D-aware mechanisms, which we clarified extensively during rebuttal.
>
> Regarding the “2D reliance” concern, our contribution is not defined by individual modules but by the 3D-grounded orchestration that constrains all semantic operations. The anchor-constrained synthesis, depth-based grounding, and multi-view geometric verification enforce physical plausibility in 3D space—capabilities that purely 2D pipelines fundamentally lack. This distinction directly addresses FRUV’s doubt about 3D specificity.
>
> On the issue of error accumulation, the reviewer was unaware of our three-layer verification design, which is purpose-built to prevent cascading errors (semantic VLM validation → multi-view consensus → depth-domain grounding). These mechanisms were detailed in the common response and resolve the primary reason cited for the low score.
>
> Given that FRUV’s critique is rooted in correctable assumptions rather than fundamental flaws, and that we have provided targeted clarifications, we believe these concerns have been fully addressed and should not be interpreted as indicators of structural weaknesses in the method.
>
> ---
>
> ## 4. Conclusion
>
> Reviewers agreed on the importance of the problem, the strong qualitative results, and the usefulness of the intent-driven formulation.
>
> The rebuttal addressed the primary concerns with substantial clarifications and the addition of missing details.
>
> We hope this helps the Area Chair make an informed decision.

---

### Meta-Review · Area_Chair_r5es · 2025-12-08

**Summary:**

Error accumulation/stability. Many VLM/segmentation steps could lead to snowballing mistakes; unclear how failures are detected (q5m4, shNF, FRUV).

Missing details. Dataset composition, initial region proposal, “improved SAGS” specifics, user-study metric definitions, some typos  (q5m4).

Quantitative evaluation. Heavy reliance on VLM judges and user studies; limited failure analysis (shNF).

Novelty & 3D specificity. Method makes heavy use of off-the-shelf 2D modules—what is the 3D contribution beyond combining them? (shNF, FRUV).

Scope & robustness. Scalability to occluded scenes, instruction diversity, handling noisy geometry, applicability beyond the chosen scene format (e.g., NeRF/Mesh) (shNF).

CLIP ambiguity & lighting. CLIP false positives from textured look-alikes; robustness relighting (rpaL).

**Reviewer Concerns:**

Error accumulation/stability: Partially addressed.
Authors describe three stages, but there’s no ablation quantifying each stage’s contribution.

Missing details: Addressed.
Added dataset numbers (20 scenes / 100 tasks), user-study metric definitions, the origin of Fig. 5 regions, typos fixed.

Quantitative evaluation: Partially addressed.
The authors argue that ground-truth doesn’t exist for open-ended editing; use HPSv2 + multi-view VLM judge. This is stronger than before but failure-mode quantification still is minimal.

Novelty & 3D specificity: Partially addressed.
The authors position the contribution as 3D-aware orchestration: intent decomposition, strategic view selection, and anchor-constrained, depth-grounded insertion. This clarifies the 3D role, but skepticism about reliance on 2D components may persist since the SAGS improvement is an engineering adaptation, not a new model, and it remains unclear whether the 3D orchestration alone drives gains.

Scope & robustness: Partially addressed.
Robustness to noisy/partial geometry is declared out-of-scope (the authors assume metric-accurate professional assets).

CLIP ambiguity & lighting: Addressed.
CLIP is a high-recall but coarse filter; ambiguities are resolved by multi-view VLM reasoning. Lighting handled via depth-based grounding (illumination-invariant).

**Reviewer Scores:**

I cannot guess how the reviewers would have changed their scores based on the subsequent discussion, of which I only see part. But three reviewers started with a score suggesting rejection. Only one reviewer was mildly positive. Not all objections have been fully eliminated.

---

### Decision · Program_Chairs · 2026-01-26

Reject